# Factors of the bone marrow microniche that support human plasma cell survival and immunoglobulin secretion

Doan C. Nguyen [1], Swetha Garimalla[2], Haopeng Xiao [3], Shuya Kyu[1], Igor Albizua[4], Jacques Galipeau[5], Kuang-Yueh Chiang[6], Edmund K. Waller[7], Ronghu Wu [3], Greg Gibson [2], James Roberson[8], Frances E. Lund[9], Troy D. Randall[10], Iñaki Sanz[11,12] & F. Eun-Hyung Lee[1,12]

Human antibody-secreting cells (ASC) in peripheral blood are found after vaccination or infection but rapidly apoptose unless they migrate to the bone marrow (BM). Yet, elements of the BM microenvironment required to sustain long-lived plasma cells (LLPC) remain elusive. Here, we identify BM factors that maintain human ASC > 50 days in vitro. The critical components of the cell-free in vitro BM mimic consist of products from primary BM mesenchymal stromal cells (MSC), a proliferation-inducing ligand (APRIL), and hypoxic conditions. Comparative analysis of protein–protein interactions between BM-MSC proteomics with differential RNA transcriptomics of blood ASC and BM LLPC identify two major survival factors, fibronectin and YWHAZ. The MSC secretome proteins and hypoxic conditions play a role in LLPC survival utilizing mechanisms that downregulate mTORC1 signaling and upregulate hypoxia signatures. In summary, we identify elements of the BM survival niche critical for maturation of blood ASC to BM LLPC.

[1] Division of Pulmonary, Allergy, Critical Care & Sleep Medicine, Emory University, Atlanta, GA, USA. [2] School of Biological Sciences, Georgia Institute of Technology, Atlanta, GA, USA. [3] School of Chemistry and Biochemistry, Georgia Institute of Technology, Atlanta, GA, USA. [4] Department of Human Genetics, Emory University, Atlanta, GA, USA. [5] Department of Medicine & University of Wisconsin Carbone Cancer Center, University of Wisconsin in Madison, Madison, WI, USA. [6] Division of Hematology & Oncology, University of Toronto, Toronto, ON, Canada. [7] Pediatrics & Hematology/Oncology, Emory University, Atlanta, GA, USA. [8] Department of Orthopedics, Emory University, Atlanta, GA, USA. [9] Department of Microbiology, University of Alabama at Birmingham, Birmingham, AL, USA. [10] Division of Clinical Immunology & Rheumatology, University of Alabama at Birmingham, Birmingham, AL, USA. [11] Division of Rheumatology, Emory University, Atlanta, GA, USA. [12] Lowance Center for Human Immunology, Emory University, Atlanta, GA, USA. These authors contributed equally: Doan C. Nguyen, Swetha Garimalla  Correspondence and requests for materials should be addressed to F.-H.L. (email: f.e.lee@emory.edu)

Human long-lived plasma cells (LLPC) that persist in the absence of antigen for decades after the original infection are the main source of protective anti-viral long-lived antibodies[1]. The differentiation and maturation programs of early circulating antibody-secreting cell (ASC) to LLPC remain elusive although the bone marrow (BM) microniche appears to play an important role. A major limitation to studying this process of human plasma cells is our inability to interrogate them in vitro. To overcome this problem, we have focused on developing an in vitro assay mimicking BM maturation to understand the survival mechanisms of human LLPC.

Early mouse and human studies have demonstrated the ability of "BM feeder cells" or BM mononuclear adherent cell (BMMC) to partially support ASC survival for 7–21 days ex vivo; however, these experiments may have contained contaminating BM ASC[2–4]. Later studies showed that ASC survival and function may be mediated at least in part by mesenchymal stromal cells (MSC) or adherent stromal cells through syngeneic cell–cell contact with plasma cells[4,5]. Representing only 0.001–0.1% of nucleated BM cells[6], MSC are adherent fibroblastoid-like cells that can be readily expanded in culture to secrete cytokines and chemokines such as IL-6 and CXCL12 which mediate the migration and retention of CXCR4[+] cells to the BM[7]. MSC also secrete critical growth factors that support hematopoiesis by providing microniches through cell–cell interactions and by establishing a rich cytokine milieu. Despite the preferential homing of ASC to the BM[8], relatively little is known about the exact cues MSC impart on ASC survival or whether the MSC play a role in the LLPC maturation process.

In addition to local stromal cells within the BM microniche, other cell types such as eosinophils[9], megakaryocytes[10], basophils[11], monocytes[12], and dendritic cells[13] have been described as potentially playing a role in ASC survival. Additionally, specific signals including ligands for the receptor BCMA (APRIL (a proliferation-inducing ligand) and possibly BAFF (a B cell-activation factor)) have also been prominently featured[14]. In mice, Mcl-1, which is an important negative regulator of apoptosis through BCMA signaling, was found to be essential for BM plasma cell survival[15]. Other cytokines and chemokines have also been implicated such as IL-5, IL-6, TNFalpha, CXCL12, and signals acting through CD44[2,11,14,16]. However, individual factors only partially support ASC for days[5] and are not sufficient for long-term survival.

Another unique characteristic of the BM microenvironment is its lower oxygen tension. Relative to other organs, the BM is naturally hypoxic[17,18] with an $O_2$ tension <10 mmHg[19–22]. However, whether hypoxia is beneficial, detrimental, or even involved in the maintenance of ASC requires further elucidation.

In this study, we tested unique features characteristic of the BM microniche to evaluate their contribution to the survival of primary human blood ASC in culture. Using IgG Elispots to measure individual plasma cell survival and function, we evaluated cellular co-cultures, cell-free secretomes of BM-MSC alone as well as in combination with exogenous cytokines under normoxic and hypoxic conditions. This approach allowed us to define a new in vitro system able to sustain human ASC for several months. In addition, we applied an integrated genomic approach matching potential protein–protein interactions identified from the MSC secretory proteome with genes that were differentially expressed between circulating human ASC and LLPC. This analysis identifies new proteins, fibronectin and YWHAZ, in the MSC secretome along with APRIL and specialized conditions (hypoxia) from the BM microniche that play a role in LLPC maturation process of survival.

## Results

**MSC co-cultures support short-term ASC survival.** We FACS purified circulating ASC (identified by CD19[+]CD38[hi]CD27[hi])

from two healthy adults, one at steady state and another 7 days after vaccination with PNEUMOVAX®23 (PPSV23). Sorted cells (1000–1500 ASC/well) were cultured in conventional media (RPMI with 10% fetal bovine serum (FBS)) or co-cultured with human BM-derived irradiated MSC (iMSC) for 7 days. Elispots were performed daily for 7 days to enumerate IgG-secreting cells. Consistent with the pronounced death rate of human ASC ex vivo, very few ASC could be detected on day one and were essentially absent by day 3 when cultured in conventional media (Fig. 1a). In contrast, >50% of the maximal ASC seeded in MSC co-cultures readily survived for 7 days. Maximal IgG ASC were determined by the peak of the Elispot responses which typically occurred on days 1–3 of the cultures.

**Long-term survival and function of ASC in MSC co-cultures.** Next we evaluated the ability of iMSC to support ASC survival in culture for extended periods of time. iMSC rather than non-irradiated MSC (noniMSC) were used for these studies to avoid MSC outgrowth, senescence, and possible consumption of essential nutrients required for ASC. Two ASC samples (one from a healthy adult at steady state and another 8 days after Tetanus, Diphtheria, Pertussis (Tdap) vaccination) were cultured at 333–1500 ASC/well with 25,000 or 30,000 iMSC/well for up to 45 days. The co-cultures were tested by IgG Elispot daily for the first 3 days and weekly thereafter. As before, a significant increase in the frequency of IgG ASC was detected within the first 72 h of culture. Approximately 20–25% of the maximal IgG ASC secreted IgG after 2 weeks in culture and 15–20% after 31 days (Fig. 1b). While this represents an improvement relative to cultures devoid of iMSC, this drop in survival suggested that replenishment of iMSC or additional factors are needed to sustain long-lived survival of ASC.

**Cell–cell contact is not required for ASC survival.** To understand if cell–cell contact is needed for plasma cell survival, 1000–2500 blood ASC obtained from two subjects after hepatitis B vaccination and 2014–2015 trivalent influenza vaccination were co-cultured directly on iMSC, with iMSC in transwells (0.4 μm pore size), or with MSC secretome alone. No significant differences in IgG Elispot frequencies were observed between direct MSC:ASC co-cultures or transwell MSC:ASC co-cultures at any time point within 6 days, thereby establishing that cell–cell contact is not necessary for ASC survival (Fig. 1c, d). As before, ASC in media alone died within one day, but a significant recovery of ASC function was observed within 24 h of co-cultures. As suggested by the transwell experiments, we tested whether secreted MSC factors were enough to support ASC survival. To do so, we isolated soluble secreted factors from MSC cultures and determined that, in isolation, the MSC secretome sustained ASC survival equal to co-cultures and transwells for 6–7 days ($p < 0.925$; linear regression modeling using media (secretome) as a covariate).

Similar to the long-term co-cultures with iMSC, the ability of the MSC secretome to support ASC survival and IgG secretion could be extended in culture for up to 56 days (Fig. 1e). Although the MSC is a relatively rare BM cell population with frequencies of 1:10,000 or 1:100,000 ([23]) compared to the BM ASC frequencies of 1:1000 ([1]), we show that the local concentration of the secreted MSC products provide the critical support for ASC. Nonetheless, the secretome similar to the co-cultures with iMSC only supports 20% of the ASC by day 30 suggesting additional factors are needed to sustain ASC long-term.

**MSC secretomes restore the secretory function of sorted ASC.** As shown in Fig. 1a, b, IgG Elispot frequencies of co-cultured

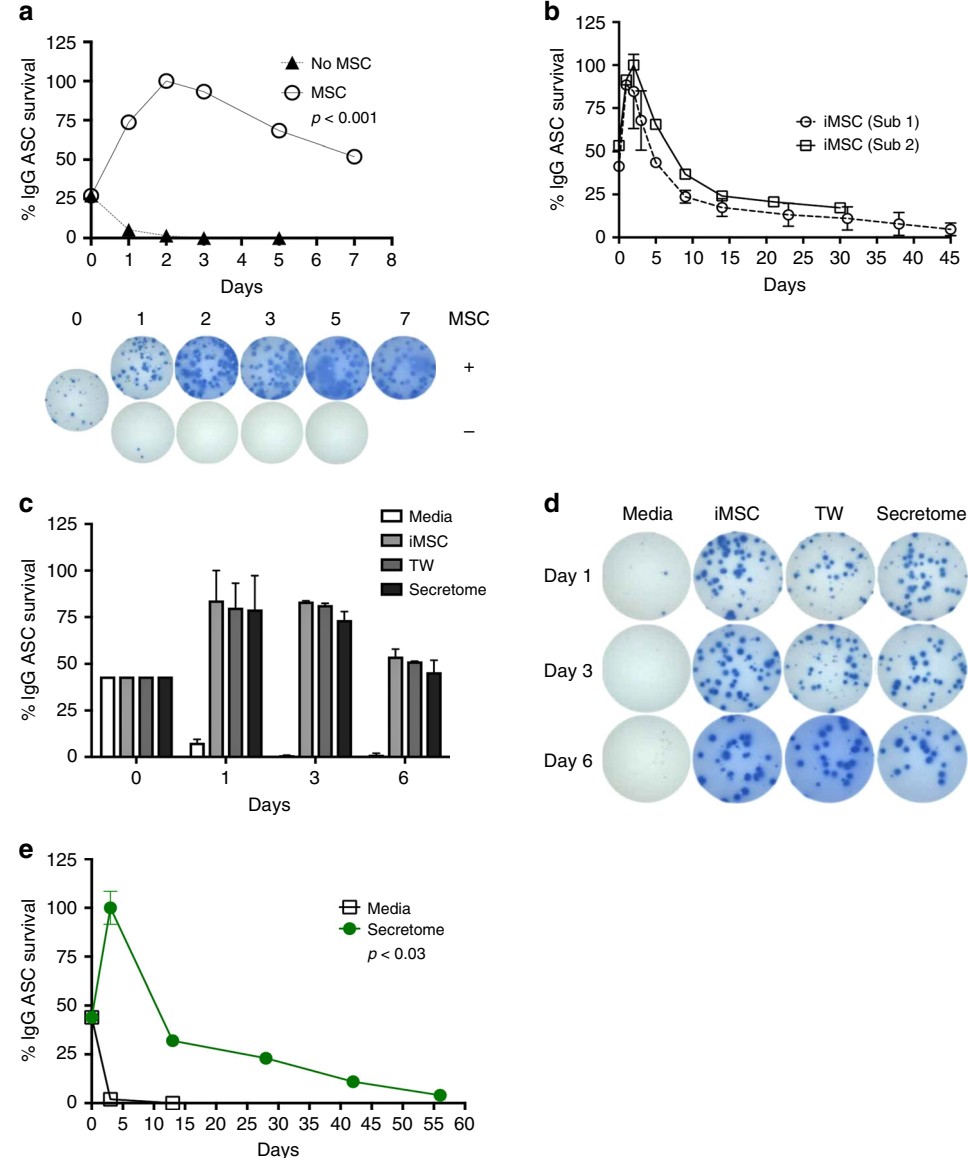

**Fig. 1** BM-MSC support in vitro survival of blood ASC. **a** Short-term survival of blood ASC in irradiated MSC (iMSC) co-cultures. FACS purified blood ASC (1000/well) were cultured in RPMI with 10% FBS (R10) (No MSC) or co-cultured with 50,000 BM-MSC (MSC) for 7 days ($p < 0.001$; ANOVA). Representative images of Elispot wells are shown. **b** Long-term survival of blood ASC and iMSC co-cultures from two different adult blood samples after vaccination. Five hundred to 1500 ASC with 25,000 or 30,000 iMSC were co-cultured in each well and IgG Elispots were performed on designated days. **c**, **d** Pro-survival support of BM-MSC is independent of cell–cell contact. Same number of blood ASC were cultured in R10 (Media), co-cultured with 50,000 iMSC (iMSC), in transwells with ASC and 50,000 iMSC in separate chambers (TW), or iMSC secretome (Secretome). Representative Elispot wells are shown in **d**. **e** Long-term survival of blood ASC in MSC secretome. One thousand ASC were cultured in MSC secretome or with media alone ($p < 0.03$; linear regression modeling using media (secretome) as a covariate). In **a**–**c**, **e**, Elispot assays were performed on the indicated day and the frequency (%) of IgG-secreting ASC were calculated based on the maximal Elispots on days 1, 2, or 3. Each figure is representative of >3 different experiments

ASC increased greatly during the first 72 h of culture compared to day 0 (representative of over 10 experiments). As previously shown, nearly all input ASC from the blood 7 days post-vaccination were Ki67[+] ([1]) (Fig. 2a) suggesting recent or ongoing proliferation of ASC. Thus, the increase in IgG ASC early in the cultures could be explained either by in vitro proliferation or restoration of IgG secretory function from non-proliferating (but recently proliferated) ASC. Because the decreased functionality immediately after sorting could be mediated by the high pressure stress of FAC sorting or the staining process, we compared Elispots of equal numbers of peripheral blood mononuclear cells (PBMC) after staining with the antibody panel (including CD27 and CD38) or leaving them untouched (no staining). In isolation,

this manipulation did not decrease the overall spot numbers (Fig. 2b). However, the impact of FACS sorting, which creates significant sheer stress, caused a nearly three-fold reduction in number of IgG Elispots compared to untouched PBMC despite the same number of total PBMC that were collected after being subjected to the high pressure FACS. As expected, the greatest reduction in functional IgG ASC was observed with a combination of staining and sorting on PBMC ($p < 0.002$; ANOVA).

To assess for ASC proliferation in the cultures, bromodeoxyuridine (BrdU) was added to the wells with secretome alone or secretome with ASC from two different subjects for 2 days. As positive controls, noniMSC which actively proliferate in culture showed a positive $OD_{450}$ value of 0.44. As negative controls, the

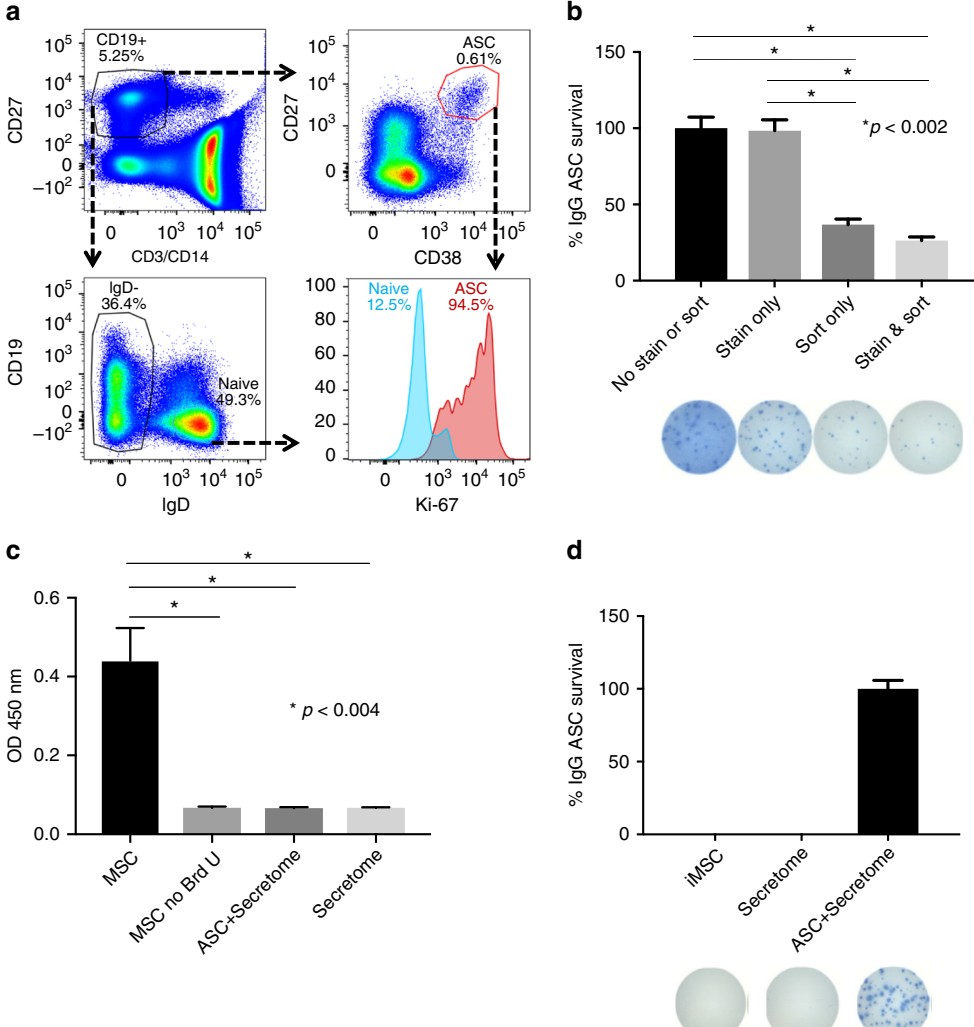

**Fig. 2** MSC secretome restores the secretory function of sorted ASC. **a** Flow cytometric analysis of intracellular Ki-67 expression of blood ASC. **b** Effects of antibody staining and sorting on ASC survival and function. IgG Elispot assays were performed on the same number of PBMC that were untouched (No Stain or Sort), only stained with antibodies and not FAC sorted (Stain), only FAC sorted and not stained with antibodies (Sort), or both stained and FAC sorted (Stain & Sort). Maximal number of IgG Elispots occurred in the No Stain or Sort wells and normalized to 100% (*$p < 0.002$; ANOVA). **c** BrdU incorporation of MSC and ASC. BrdU was added to ASC cultured in the secretome or secretome alone. Triplicate cultures of dividing noniMSC alone with and without BrdU served as positive and negative controls. BrdU was also added to conditions of secretome alone or secretome with ASC (*$p < 0.004$; ANOVA). This figure is representative of three experiments. **d** No BM-derived ASC contamination in iMSC co-cultures or MSC secretome. IgG Elispot assays were performed on iMSC (iMSC) or MSC secretome (Secretome). Blood ASC in MSC secretome cultures (ASC + Secretome) served as a positive control. Representative images of Elispot wells are shown below

OD$_{450}$ of <0.1 was measured in MSC without BrdU added (Fig. 2c). When tested, the BM-MSC secretome alone was negative for BrdU uptake which was not surprising since filtering through a 0.2 um filter would have removed any remaining cells. However, surprisingly, the ASC incubated with the secretome was also negative for BrdU uptake demonstrating that the human ASC found in circulation after vaccination are not actively dividing in the cultures despite positive Ki67 staining showing cells have undergone recent proliferation but are not actively proliferating ($p < 0.004$; ANOVA). In conclusion, ASC survival and IgG secretory function are diminished by FACS sorting and recover from a stunned non-secretory state without further proliferate within 48 h in our culture conditions.

To ensure that the iMSC or the secretome did not contain contaminating plasma cells derived from the BM, we performed IgG Elispots from the iMSC cultures and their secretome alone. No IgG Elispots were detected in the iMSC alone or the MSC secretome cultures (Fig. 2d). We only detected IgG ASC in the cultures with the secretome together with sorted blood ASC demonstrating IgG Elispots originated from input blood ASC and not from any contaminating BM plasma cells carried over with the iMSC or the MSC secretome preparations.

Since ASC were not proliferating in the cultures and the numbers of viable IgG ASC varied with each sample and FAC sort, we normalized each experiment to the maximal IgG Elispot numbers detected on days 1–3. On day 0, approximately 3–30% of input cells were able to secrete IgG antibodies. This number may be low due to several reasons: overestimates of cell counts after FAC sorting, apoptosis due to sheer stress of cells after collection, or loss with washing steps. Additionally, from blood CD19$^{lo}$CD27$^{hi}$CD38$^{hi}$ sorted cells, a significant percentage of IgA circulating ASC can account for nonIgG secretors that could

range from 10% to 60%. In order to study the long-term survival and function of ASC in BM maturation conditions, we measured percentage of subsequent ASC survival beyond day 1.

**APRIL enhances ASC survival induced by the MSC secretome**. TNF-ligand superfamily cytokines (APRIL and BAFF) are known to have a seminal role in the survival of B lymphocytes and plasma cells[14,24]; but APRIL has been implicated most strongly in the generation of LLPC[9,14]. We had previously shown that B cell maturation antigen (BMCA), a TNF superfamily (TNFRS17) receptor for APRIL, is highly expressed by all human BM ASC including the LLPC[1]. Thus, we tested the effect of APRIL alone or in combination with the MSC secretome on the survival of ASC. As shown in Fig. 3a, b, exogenous APRIL alone provided no ASC survival advantage over conventional media. A similar lack of benefit was observed with other individual cytokines such as IL-5, IL-6, and BAFF alone (Supplementary Figure 1).

Despite APRIL providing no survival advantage when used in isolation, the addition of exogenous APRIL to the MSC secretome resulted in more than two-fold enhancement in functional IgG ASC frequency over the first 24 h of culture followed by significantly enhanced ASC survival over 7 days ($p < 10^{-6}$; linear regression modeling using media (secretome or secretome + APRIL) as a covariate) (Fig. 3a, b). This increase was significantly greater than ASC cultured with the MSC secretome alone from days 1 to 7. This phenomenon was statistically significant in four additional subjects after vaccines including tetanus, influenza virus, and measles & mumps-rubella (MMR) (Fig. 3c). Interestingly, the addition of APRIL to the MSC secretome promoted long-lasting ASC survival (14 days and beyond) at rates significantly superior to the activity of the MSC secretome alone. This additional survival benefit of exogenous APRIL was sustained up to 56 days which trended towards significance ($p < 0.09$; linear regression modeling using media (secretome or secretome + APRIL) as a covariate) (Fig. 3d). In some samples, the slopes of ASC survival of the secretome alone vs secretome and APRIL were different; however, it did not reach statistical significance possibly due to small samples sizes or a survival advantage of APRIL during the recovery phase. Thus, the presence of APRIL in the MSC secretome enhanced short-term (days) and likely long-term (weeks) ASC survival.

In contrast to APRIL, the addition of BAFF to the MSC secretome provided no additional survival benefit (Supplementary Figure 2). Furthermore, the addition of IL-6, IL-5, IFNg, VEGF, or basic fibroblast growth factor (bFGF or FGF2) to the MSC secretome also offered no further enhancement of survival (Supplementary Figure 2). Despite additional IL-6 conferring no benefit, inhibition with IL-6 antibodies in the secretome significantly diminished survival (Fig. 3e) on days 1, 3, and 7 compared to MSC secretome alone ($p < 10^{-5}$; linear regression modeling using media (secretome) as a covariate) demonstrating that the secretome contained abundant amounts of IL-6 as we and others had previously shown[25].

**Proteomics of the MSC secretome**. While IL-6 is known to be secreted by the MSC, to identify novel MSC plasma cell survival factors we utilized a combined bioinformatic approach of proteomics and transcriptomics (Fig. 4a). Initially, we performed proteomics of MSC secretome fractions with and without ASC survival activity. We isolated secretomes from irradiated and non-irradiated MSC and fractionated the supernatants by ultracentrifugation. Additionally, secretomes from blood (not BM) adherent cells (SBAC) were also generated. The fractions with biological activity included the irradiated or non-irradiated MSC secretomes and the irradiated

supernatant, whereas the non-irradiated supernatant and the SBAC showed decreased ASC survival (Fig. 4b). Proteomics was performed on irradiated and non-irradiated secretomes and irradiated and non-irradiated supernatants along with the SBAC fraction, and peptide fragments were aligned using DAVID. Background of FBS was subtracted. The union of the positive biologic fractions (irradiated and non-irradiated secretomes together with the irradiated supernatant) yielded 231 proteins. One hundred fifty-six proteins were identified from the irradiated MSC secretome proteome subtracting the SBAC proteome. The number of overlapping proteins between the two experiments was 91. Then, we employed HIPPIE[26] to discover 4426 potential protein–protein interactions (PPI) with the 91 MSC proteins.

To further narrow the MSC PPI that may play a role in ASC survival process, we turned to the differentially expressed genes (DEG) between the early minted ASC in the blood compared to the LLPC in the BM as previously described (SRA:SRP057017)[1]. Transcriptomes of 17 new blood ASC samples using the same surface markers described in BM ASC subsets (pop A, B, and pop D (LLPC))[1] and the four BM LLPC (pop D) were aligned, normalized, and analyzed with standard statistical methods. We found a total of 2558 genes were differentially expressed between blood ASC and BM LLPC at a false discovery rate (FDR) of 0.05 (Fig. 4c). Using an integrated approach designed to assess the role of the MSC proteome on the LLPC maturation, we overlapped the 4426 potential PPI from the MSC proteome with the 2558 DEG between blood ASC and BM LLPC transcriptomes and identified 556 overlapping gene/protein targets (Supplementary Data 1). Gene set enrichment analysis (GSEA)[27–29] revealed 20 statistically significant Hallmark pathways (Fig. 4d and Supplementary Data 2). Highlighted pathways included mTORC1 signaling, PI3K–Akt–mTOR signaling, TNFalpha signaling, and hypoxia.

We also noticed that 12 of the 91 proteins that had more than 45 PPI that overlapped with the DEG profiles of blood ASC to BM LLPC (Fig. 4e). The top proteins consisted of fibronectin (FN-1), YWHAZ (also known as 14-3-3zeta/delta), heat-shock proteins, enolase, glyceraldehyde 3-phosphate dehydrogenase (GAPDH), eukaryotic elongation factor (EEF2), and various cytoskeletal proteins (actin, tubulin, and vimentin). The top two proteins, FN-1 and YWHAZ, had 436 and 430 PPI, respectively, of which 118 and 119 overlapped with the DEG in blood ASC and LLPC (FDR < 0.05; $p < 0.05$; ANOVA; Supplementary Data 3). The pathways involved for only FN-1 and YWHAZ involved in only 10 of the 20 GSEA hallmark pathways (Fig. 4f and Supplementary Data 4). These pathways included proliferation signatures, i.e. E2F targets and G2M checkpoints which is consistent with evidence that most blood ASC have undergone recent proliferation (Fig. 2a) in contrast to the BM LLPC[1]. However, the downregulation of Myc targets, mTORC1 signaling, and PI3K–Akt–MTOR signaling suggested novel pathways that the MSC secretome may be involved in maintaining ASC survival.

To validate the importance of the top two proteins, we inhibited FN-1 and YWHAZ with anti-fibronectin and anti-YWHAZ antibodies in the MSC secretome and showed statistically significant decrease in ASC survival after 1, 3, and 7 days compared to no inhibition or isotype controls ($p < 10^{-3}$ and $< 10^{-5}$, respectively; linear regression modeling using media (secretome) as a covariate) (Fig. 5a, b). In summary, we identified 91 unique potential proteins from the MSC secretome that may play a role in ASC survival. FN-1 and YWHAZ had the strongest PPI and their inhibition resulted in decreased ASC survival. Whether the other 89 proteins have a direct role in LLPC maturation will require further investigation.

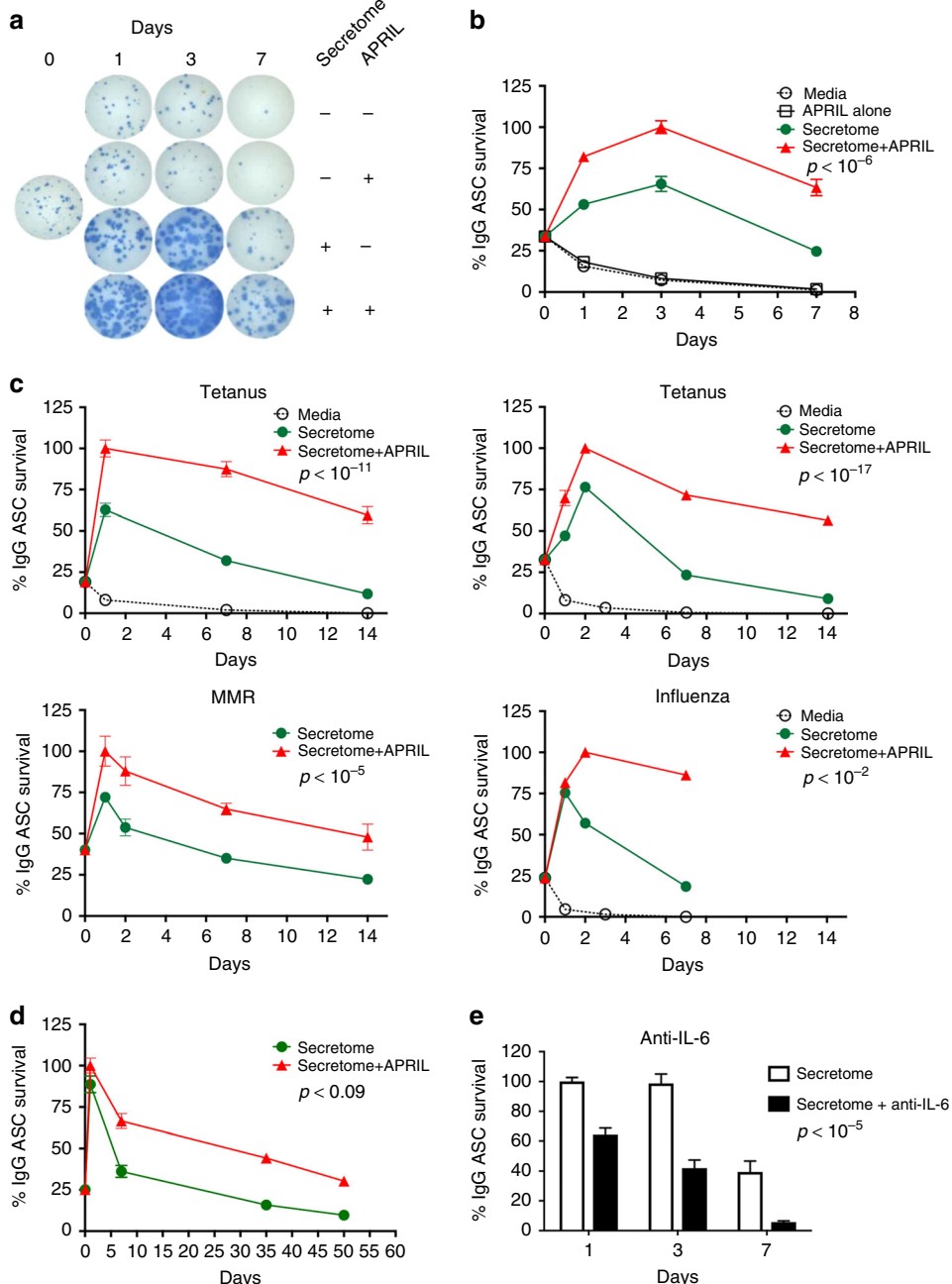

**Fig. 3** APRIL together with the MSC secretome enhances blood ASC survival. **a, b** Blood ASC survival is enhanced by the MSC secretome with exogenous APRIL. First two rows: sorted blood ASC were cultured in R10 alone or in media with exogenous APRIL for 7 days. Second two rows: blood ASC were cultured in MSC secretome alone or the secretome with APRIL. Blood ASC immediately from FAC sorting are shown (day 0). Representative images of Elispot wells are shown. Graphic plot of % survival of IgG ASC in media alone (R10) (open circles), APRIL alone (open squares), secretome alone (green circles), or secretome + APRIL (red triangles) ($p < 10^{-6}$; liner regression modeling between secretome alone and secretome + APRIL). **c** Exogenous APRIL together with the MSC secretome enhances ASC survival in short-term cultures. Blood ASC from four different subjects after vaccination were cultured in media alone (open circles), secretome alone (green circles), or secretome + APRIL (red triangles). Percentage of IgG Elispots normalized to maximal frequency on days 1–3. Shown are $p$-values between secretome alone & secretome + APRIL. **d** Exogenous APRIL with the MSC secretome enhances ASC survival in long-term cultures. ASC cultured with the MSC secretome (green circles) or MSC secretome + APRIL (red triangles). **e** Treatment of MSC secretome with anti-IL-6 antibodies (anti-IL-6) diminishes blood ASC survival. Blood ASC were cultured in MSC secretome or MSC secretome treated with anti-IL-6 for 1, 3, 7 days and IgG Elispots were performed ($p < 10^{-5}$; linear regression modeling between secretome vs secretome + anti-IL-6). The frequency of isotype controls was similar to MSC secretome. Representative of three experiments

**Role of mTORC1 signaling in LLPC maturation.** Our integrated proteomics and transcriptomics posited a role for the MSC secretome in downregulating mTORC1 signaling as early blood ASC enter the BM microniche and mature into LLPC. Thus, we hypothesized that early minted blood ASC would be sensitive to Rapamycin, an mTORC1 inhibitor, while BM LLPC would be resistant to Rapamycin. Indeed, blood ASC showed decreased survival with Rapamycin validating enhanced mTORC1 signaling in the blood ASC from three subjects (Fig. 5c). In contrast, BM LLPC from three adult BM samples were entirely resistant to

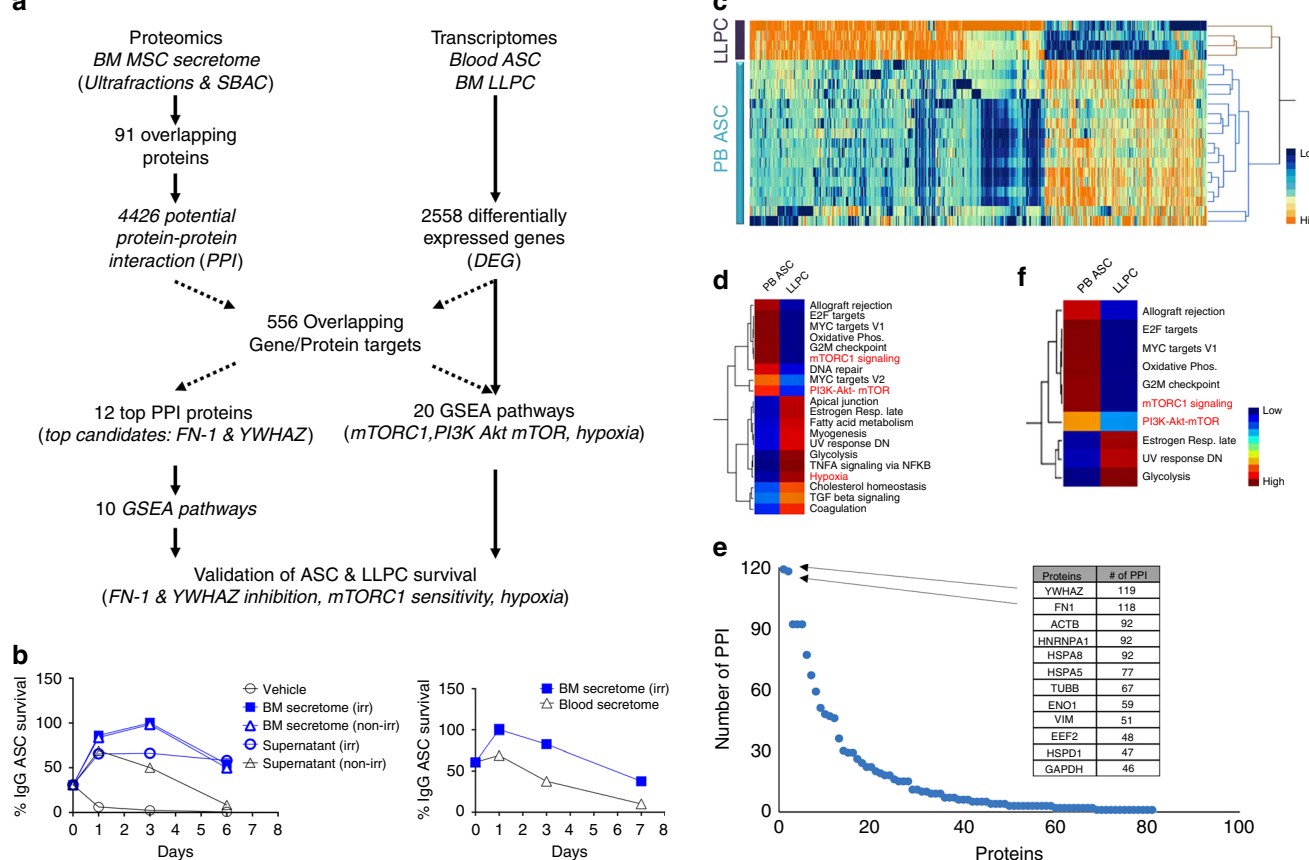

**Fig. 4** Integrated bioinformatics of MSC secretome proteomics with transcriptomics of blood ASC and BM LLPC. **a** Flow diagram of the integrated bioinformatic analysis. **b** Proteomics of distinct MSC secretome fractions. Three fractions strongly support blood ASC survival (blue symbols): iMSC secretome (blue filled square; left and right panels), noniMSC secretome (blue open triangles; left panel), and supernatant fractionated by ultracentrifugation of iMSC secretome (blue open circles; left panel); and three fractions with decreased ASC survival (black symbols): conventional media (R10) (black open circles; left panel), supernatant fractionated by overnight ultracentrifugation of noniMSC secretome (black open triangles; left panel), and secretome from blood (not BM) adherent cells (black open triangles; right panel). **c** A heat map of the 2558 DEG between 17 blood ASC (PB ASC) obtained from seven healthy subjects and BM LLPC (LLPC) obtained from four adult subjects. **d** HIPPIE analysis of 91 MSC protein revealed 4429 potential protein partners (PPI). Overlap of the 4429 PPI with the 2558 DEG uncovered 556 overlapping gene/protein targets and led to 20 statistically significant GSEA hallmark pathways. **e** Of the aforementioned 91 MSC secretome proteins, FN-1 and YWHAZ had the highest number (118 and 119, respectively) of potential interacting partners. Of these potential partners, 31 were shared between both FN-1 and YWHAZ. **f** From the 20 GSEA pathways, FN-1 and YWHAZ were found to be involved in these 10 potential GSEA hallmark pathways

Rapamycin confirming the downregulation of mTORC1 signaling in LLPC (Fig. 5d). These results highlight the importance of proteins from the MSC secretome such as FN-1 and YWHAZ and their putative role in downregulating mTORC1 signaling as early minted blood ASC mature into LLPC.

**ASC survival is enhanced under hypoxic conditions**. Of the 20 GSEA hallmark pathways discovered with the integrated bioinformatics of the MSC proteome and DEG of the blood-BM transcriptomes, hypoxia hallmark pathway was prominently featured. Interestingly, the hypoxia pathway is not enriched for interactions involving FN-1 and YWHAZ, suggesting that upregulation of hypoxia signatures are independent of these top proteins (Fig. 4c, d). To investigate the role of hypoxia in ASC survival, we isolated circulating ASC 7 days after tetanus immunization and measured ASC survival in the following three conditions in normoxia and hypoxia: conventional media alone, MSC secretome, and MSC secretome with APRIL (Fig. 6a, b). In keeping with the previously described results (Fig. 1b, e), conventional media in normoxic or hypoxic conditions provided no support. Similar to previous results, the MSC secretome alone under normoxia showed only moderate ASC survival (9% ± 1%)

by day 14, and under hypoxia, the same conditions improved ASC survival at day 14 (18 ± 2%). As previously shown, the MSC secretome with APRIL improved survival particularly beyond 14–56 days in normoxic conditions (56% and 19%, respectively). However, hypoxic conditions in combination of APRIL and the MSC secretome further enhanced ASC survival to over 66% at day 14 and over 40% at day 56. Thus, in this context, hypoxia significantly enhanced ASC survival in the MSC secretome and APRIL combined at every time point from day 14 to 56 ($p = 10^{-26}$; linear regression modeling using media (secretome or secretome + APRIL) as a covariate). Together, our results demonstrate the prominent synergistic effect on ASC survival with hypoxia in combination with the MSC secretome and APRIL, suggesting that BM environmental factors play significant roles in novel mechanisms of LLPC maturation leading to prolonged survival.

**Discussion**

Plasma cells are the main source of both protective microbial antibodies and pathogenic autoantibodies and represent the effector arm of the humoral system. From a cellular standpoint, ASC are a heterogeneous compartment comprised of multiple

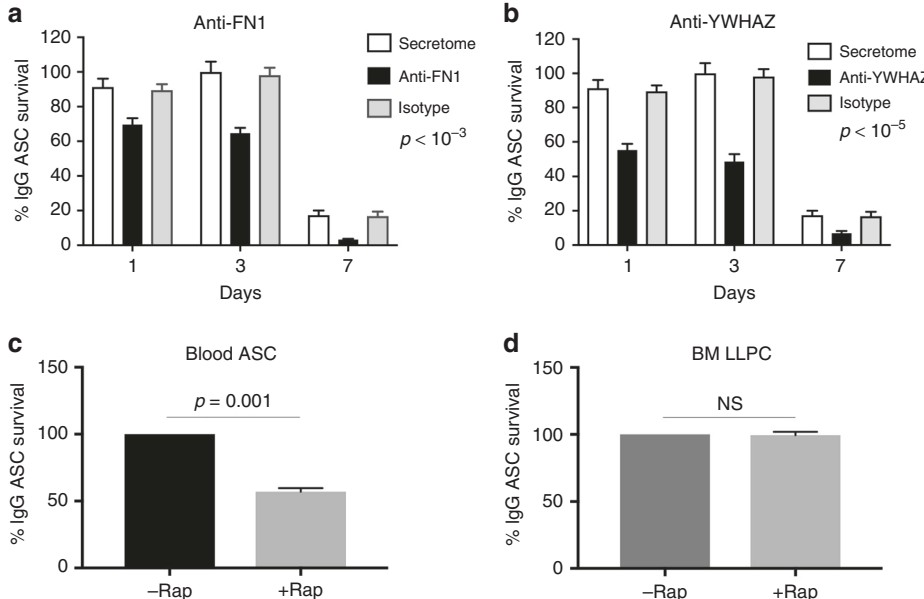

**Fig. 5** Validation of ASC survival. Treatment of MSC secretome with antibodies targeting **a** FN-1 or **b** YWHAZ diminishes blood ASC survival. Blood ASC were cultured in MSC secretome alone (white bar) or MSC secretome treated with antibodies targeting FN-1 (**a**; black bar) ($p < 10^{-3}$; linear regression modeling using media (secretome) as a covariate) or YWHAZ (**b**; black bar) ($p < 10^{-5}$; linear regression modeling using media (secretome) as a covariate), or appropriate isotype controls (gray bar). Graph represents average of at least three experiments. Sensitivity and resistance to Rapamycin in **c** blood ASC and **d** BM LLPC. Blood ASC from three healthy subjects or BM LLPC from three adults were sorted and cultured in untreated MSC secretome (black bar) or MSC secretome treated with rapamycin (gray bar). The frequency of vehicle controls (DMSO) was comparable to that of untreated MSC secretome. $p = 0.001$; ANOVA) (**c**); NS, non-significance (**d**)

subsets with distinct surface phenotypes that participate in short-lived and long-lived antibody responses. Long-lived antibodies provide persistent serological memory and account for protection against past pathogens but are also responsible for the persistence of pathogenic auto- and allo-antibodies even after therapeutic B cell depletion. Therefore, a precise understanding of the LLPC generation from B cell differentiation to BM maturation bears substantial implications for modulating human antibody responses in health and disease. However, the fragility of ASC ex vivo and their accelerated spontaneous in vitro cell death under conventional culture conditions have precluded a systematic investigation of human ASC.

We have addressed these challenges by establishing a novel in vitro cell-free human plasma cell culture system promoted by factors that mimic the BM microenvironment. These factors appear to naturally support long-term survival of ASC in vivo. While the importance of BM residence for ASC survival is well established[2,4], the particular cell types and specific mechanisms underlying ASC survival in the BM microniche have remained elusive. In this paper, we demonstrate that factors secreted by the MSC, APRIL, and hypoxic conditions constitute a major component of the plasma cell survival niche. In addition to providing a useful platform for ASC cultures, our parsimonious approach also contributes original insights into critical factors responsible for the mechanisms of LLPC maturation and survival.

BM-MSC are an essential component of the in vitro survival system and we found that their effects are mediated through a soluble secretome in the absence of cell–cell contact. Interestingly, only IL-6 has been shown to be critical in mouse models while other factors such as IL-5, TNFalpha, and CXCL12 have been suggested but not definitively shown to play a significant role in ASC survival[2,5]. Here, we identified 91 proteins in the MSC secretome that are likely to play a role in the human BM LLPC microniche. We validated functional roles for the top two candidates, fibronectin (FN-1) and YWHAZ, which were predicted

to have the highest number of protein–protein interactions. Although proteomics of the BM-MSC secretome has been previously described[25], we employed a novel approach to identify the potential protein partners that overlap with differentially expressed genes as blood ASC mature into BM LLPC phenotypes to discover FN-1 and YWHAZ as the top two of these 91 unique proteins.

FN-1 is a large glycoprotein of the extracellular matrix (ECM) that binds integrins, collagen, and heparin sulfate glycoproteins such as syndecan (CD138) a prominent surface marker of LLPC. In addition to CXCR4–CXCL12 interactions that have been shown to support homing of plasma cells to the BM microniche[7], FN-1 may also play an important role in tethering the plasma cells to the BM ECM through its interactions with integrins such as VLA-4[4,5]. Perhaps a survival advantage ensues with this particular interaction resulting in the increased number of CD138+ plasma cells in the BM location. Recently, it was reported that CD138 mediates selection of mature plasma cells and enhanced BM survival in mice[30] but the mechanisms were not clear. In our studies, inhibition of fibronectin affected human ASC survival (Fig. 4e), which in part may be mediated by direct FN-1 and CD138 interactions. In addition to binding to the ECM of the BM microniche, FN-1 may also mediate other important mechanisms such as mTOR signaling and metabolism in plasma cells. Recent studies in mouse hepatocytes showed fibronectin deficiency impairs mTOR-mediated autophagy and increased steatohepatitis[31]. Our data would suggest FN-1 may also play a role in mTOR-mediated autophagy of human BM LLPC since we observed a downregulation of mTORC1 signaling from blood to BM LLPC subsets (Fig. 4b). These results are consistent with our previous finding of increased autophagosome numbers in BM LLPC compared to BM SLPC[1]. Thus, the role of FN-1 may be multifactorial, which include the ability to tether CD138+ ASC to the hypoxic BM microniche and possibly increase autophagy programs.

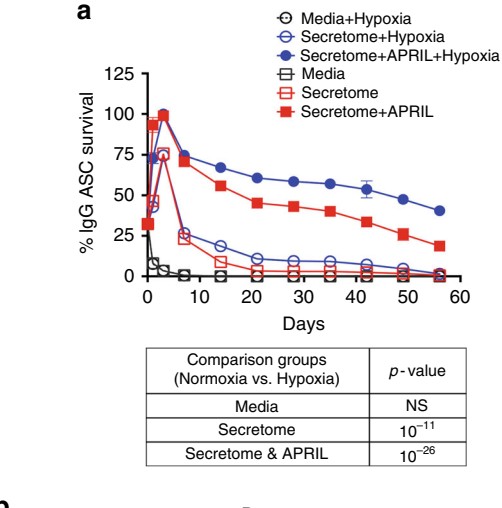

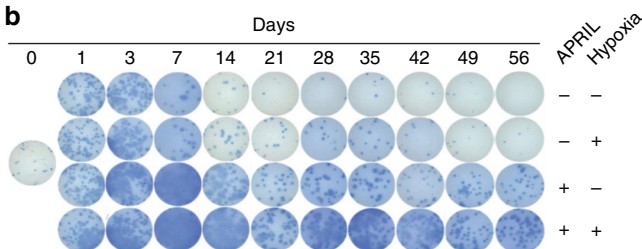

**Fig. 6** Survival of blood ASC is enhanced under hypoxic conditions with the MSC secretome and APRIL. **a** Circulating ASC post-tetanus vaccination from one healthy adult were cultured under the following conditions in normoxia: media alone (black open squares), secretome alone (open red squares), or secretome + APRIL (red closed squares) and under hypoxia conditions: media alone (black open circles), secretome alone (blue circles), or secretome + APRIL (blue closed circles). Percentage of IgG Elispots were normalized to the maximal frequency on day 3. Data are representative of four different blood ASC from four adults after vaccination (p-values shown in the table below). **b** Representative images of Elispot wells from the experiments in **a** are shown. Number of ASC per well: 1333 except on wells from days 21–56 for MSC secretome + APRIL in normoxia or hypoxia (800) to show images in a countable range

Similar to fibronectin, we show a decrease in ASC survival with YWHAZ inhibition in our BM mimic cultures. YWHAZ or 14-3-3zeta is an intracellular protein in the 14-3-3 family of proteins which are ubiquitously expressed and mediate binding to phosphoserine/threonine proteins. These proteins are involved in signaling pathways that modulate cellular metabolism and have been shown to modulate insulin signaling, mTOR, and AMP dependent kinase signaling as well as autophagy pathways[32]. YWHAZ is also known to negatively regulate apoptosis by binding and sequestering BAD and BAX in the cytoplasm, and thus preventing activation of proapoptotic BCL-2 and BCL-XL and preventing NOXA from inhibiting anti-apoptotic MCL-1 ([33]). In this study, we found that this protein appears to play an important role in ASC survival; however, it was puzzling how an intracellular proteins could be so prominent in the extracellular MSC secretome.

The 14-3-3 proteins do not possess signal peptides for secretion, and so their presence in the extracellular environment was questioned until it was shown that exosomes and extracellular vesicles (EVs) contain these proteins and could be transported from cell to cell[34,35]. Externalizaton of free 14-3-3 proteins also occurs during cell death[32], and in our secretomes from irradiated MSC, cell death may ensue during the 7 day collection period. Therefore, in comparison of the irradiated and non-irradiated

secretomes, abundance of YWHAZ in irradiated secretomes may be one reason that the irradiated supernatants afforded better ASC survival than non-irradiated supernatants (Fig. 4b). Our recent studies demonstrate an important role of EVs to mediate ASC survival which may be in part due to YWHAZ[36]. Therefore, YWHAZ may be transported via EVs or released into the media by our methods of irradiating MSC to generate the secretomes.

Another issue was understanding how the YWHAZ protein in the secretomes was internalized. YWHAZ can bind to the IL-9alpha receptor[37] and to CD13[38], a receptor known to be upregulated on human plasma cells[39]. In summary, our integrated bioinformatics approach identified a novel intracellular protein, YWHAZ, from irradiated MSC secretomes, and we show it plays a prominent role in ASC survival which may be involved in metabolic pathways such as insulin, as well as mTORC1 signaling, and autophagy.

The MSC plasma cell survival secretome is a complex mix of many factors of which mass spectrometry identified only the most highly abundant proteins. Highly sensitive cytokine arrays are needed to detect low abundant proteins such as IL-6 and CXCL12 ([25]). IL-6 was easily detected in our irradiated secretomes by ELISA (range approximately 300–2000 pg/mL), but due to its relatively low abundance in the MSC secretome, it was not identified in the proteomics analysis. Although additional exogenous IL-6 conferred no survival advantage, inhibition of IL-6 diminished ASC survival validating the prominent role of IL-6 for ASC survival (Fig. 3e). Whether IL-6 signaling is important to sustain blood ASC and LLPC or IL-6 is critical for maturation to a LLPC phenotype will require further exploration.

Our results showed how the factors of the MSC secretome can affect the LLPC maturation process. In addition to mTORC1 signaling through FN-1 and YWHAZ, the decrease of Myc targets is likely to be among the other possible LLPC maturation mechanisms (Fig. 4d). One of the top 12 MSC secretome proteins was ENO1, a glycolytic enzyme that was previously identified as a myc-binding protein (MBP-1) demonstrating that ENO1 downregulates c-myc[40]. Interestingly, we also observed downregulation of c-Myc signaling in the LLPC (BM pop D)[1]. Thus, it is possible that ENO1 will play a role in LLPC maturation. The interplay of the different survival factors may in fact have redundant function since inhibition of FN-1, FYWHAZ, and IL-6 have greater than 100% loss of survival. Therefore, understanding the overlapping functions will be important. Ultimately, studies of reconstitution and inhibition with YWHAZ and IL-6 in combination with FN-1, ENO1, as well as the other top protein candidates, GAPDH, HSPA5, HSPA8, and HSPD1 would be needed to fully characterize the MSC survival-promoting secretome.

Previous studies have demonstrated an important role for APRIL in ASC survival. In keeping with this phenomenon, APRIL imparts an enormous survival advantage when added to the MSC secretome. The APRIL-enhanced survival was observed for at least 56 days, a timeframe that strongly suggests its importance in the maintenance of human LLPC. Interestingly, APRIL alone is not beneficial suggesting that it acts synergistically with other factors in the MSC secretome or with the heparin sulfate proteoglycan which is a APRIL binding ligand[41]. Recently it was shown that APRIL increases BCL-xL expression by a preferential binding to heparin sulfate glycoproteins at the surface of CD138 ([42]), suggesting the importance of CD138 and APRIL interactions. Additionally, it was shown that BCMA is needed for MCL-1 expression in BM plasma cells[15] and that it is critical for its survival. However, in our culture system, synergy of other MSC secreted factors such as IL-6, FN-1, and YWHAZ in addition to APRIL are necessary for ASC survival. Although APRIL

secretion by MSC has been reported in mice[42], APRIL is not physiologically produced by our BM-MSC after multiple passages (our unpublished results) and in vivo APRIL is likely to be supplemented from other local sources such as eosinophils, basophils, or neutrophils.

Our study also provides strong evidence for a survival benefit induced by hypoxic conditions that recapitulate the distinct BM microenvironment. Hypoxia in media alone afforded no survival advantage. However, hypoxia enhanced ASC survival above and beyond the benefit imparted by the addition of APRIL to the MSC secretome when added together. In contrast to other conditions tested, the effects of hypoxia were noted after 7 days in culture, a time frame that suggests the engagement of adaptation programs critical for maturation mechanisms. The nature of these programs represents a central concept that the local BM-MSC paracrine factors facilitate ongoing LLPC maturation. Fortunately, our cell-free system is poised to make this maturation possible and future intervention studies will explore the role of the other factors.

In addition to its biological significance, our system provides a powerful and innovative technological platform for the interrogation of human ASC. At present, there is no known in vitro system to sustain ex vivo sorted human ASC in short- or long-term cultures. One study used irradiated PBMC together with unsorted cells from intestinal biopsies to support long-term intestinal plasma cell survival for IgA and IgM but not IgG[43]. Unfortunately, these cultures contained ASC with contaminating input B cells which did not distinguish in vitro generation of new ASC from survival of input ASC. Another system used tonsillar stromal cells but only showed short-term survival of in vitro differentiated B cell cultures[44] and not freshly isolated in vivo differentiated human ASC as shown here. Therefore, prior to this study, no cell-free system was available to support human primary ASC. Fortunately, this unique system is simple, cell-free, and robust.

In summary, this study sought to address the role of the BM microniche in maintaining blood ASC survival and also discovered a cell-free reproducible in vitro plasma cell survival system. We demonstrate that the survival of circulating early minted ASC requires a hypoxic environment with paracrine survival factors from BM stroma cells and exogenous APRIL. Two novel protein factors from the MSC secretome include fibronectin and YWHAZ which play important roles for ASC survival and probably LLPC maturation. In sum, this novel in vitro human plasma cell survival system provides basic tools to examine cellular compartments of protective and pathogenic antibodies, to interrogate antibody repertoires at a single ASC and memory B cell level, and to ultimately understand mechanisms of human plasma cell differentiation and maintenance.

## Methods

**Human subjects.** We recruited 68 adults for peripheral blood (PBL) samples ($n = 55$), BM aspirates (BMA) ($n = 9$), and discarded femoral heads (FH) ($n = 4$). All studies were approved by the Emory University Institutional Review Board Committee and informed consent was provided. PBL samples were obtained from 20 healthy subjects and 35 adults who received one or more routine vaccines at 6, 7, 8, 11, or 12 days after immunization (21–59 years old; mean ± SD, 36 ± 11). One adult (31 years old) had PCR-proven influenza virus infection. BM specimens were collected by aspiration from 9 healthy adults (22–55 years old; mean ± SD, 37 ± 14). FH samples were collected from adults who were diagnosed with joint diseases (predominantly osteoarthritis and degenerative joint disease) and underwent hip replacement surgeries (42–73 years old; mean ± SD, 63 ± 14). Sample collection dates ranged from July 2014 to April 2018.

**BM-derived mesenchymal stromal cells and secretome.** The source of BM-MSC was BMA of healthy donors. Isolation and expansion of MSC were performed as previously described[36]. Briefly, BM mononuclear cells (MNC) were isolated by Ficoll-Hypaque (GE Healthcare) or Lymphocyte Separation Medium (LSM;

Cellgro/Corning) density-gradient centrifugation[45]. BM MNC were placed in 75-cm$^2$ cell culture flasks (BD Biosciences) at ~10–20 × 10$^6$ cells per flask (or at a cell density of ~5–10 × 10$^5$ cells/mL) in 15–20 mL MSC medium (HyClone Dulbecco's Modified Eagle Medium (DMEM; GE Healthcare) supplemented with 10% heat-inactivated FBS (Sigma) and 1% Antibiotic–Antimycotic (i.e. 100 units/mL penicillin, 100 μg/mL streptomycin, and 0.25 μg/mL Fungizone (amphotericin B); Thermo Fisher)). During the following 2–7 days of cultures, non-adherent cells were removed by replenishing with the MSC medium (which contained the non-adherent cells). The plastic-adherent (stromal-like) cells at ~80–90% confluence were trypsinized and passaged. Adherent cells were subjected to flow analysis for MSC cell surface markers[46] using CytoFlex cytometer (Beckman Coulter); these cells appeared as pure populations of CD90+CD73+CD45-CD19− and were classified as primary BM-MSC. These MSC were further propagated generally between their third and eight passages. Irradiated BM-MSC (iMSC) were obtained by exposing the cells to γ-irradiation for a total dose of ~30.64 Gy.

MSC secretomes were essentially supernatants harvested from BM-MSC cultures[36]. Briefly, supernatants were harvested daily for an entire week from >80% confluent non-irradiated BM-MSC and irradiated BM-MSC monolayer cultures. Supernatants collected from all days were then pooled and centrifuged at ~500 × g for 10 min at 4 °C and then at 2000 × g for 30 min at 4 °C to remove cell debris. After filtering (0.22 μm Filter System; Corning), supernatants were subjected to sterility testing and aliquoted for immediate uses or stored at −80 °C for subsequent usage. Similar procedures were applied to obtained blood adherent cells from PBMC and to subsequently generate their secretome (SBAC). An SBAC sample (~1–2 mL) was tested for sterility before use.

**Isolation of PBMC and FACS-based purification of blood ASC.** PBMC isolation and blood ASC purification were performed[1,36,47,48]. Briefly, PBMC were separated from freshly collected PBL samples by Ficoll-Hypaque (GE Healthcare) or Lymphocyte Separation Medium (LSM; Cellgro/Corning) density-gradient centrifugation[49]. T cells and monocytes were removed by CD3 and CD14 beads (Miltenyi) and flow through stained with the following panel: human CD3-PE-Cy5.5 (or human CD3-BV711), human CD14-PE-Cy5.5 (or human CD14-BV711) (Life Tech); human CD19-PE-Cy7, human IgD-FITC, human CD27-APC-eFluor780 (or human CD27-PE), human CD38-v450, and human CD138-APC (BD Biosciences). After washing, PBMC were sorted on the FACSAria II sorter (BD Biosciences). The resultant blood ASC populations (CD19+CD27hiCD38hi) were generally ~85–99% pure, as assessed by flow cytometric re-analysis of post-sort cells. Post-sort ASC were subjected to cultures immediately.

**In vitro culture systems for human blood and BM ASC.** BM-MSCs as feeder (co-cultures on adherent monolayer) were co-cultured in 96-well flat-bottom cell culture plates or transwells (0.4 μm pore polycarbonate insert membrane of 96-well plates (Corning/Sigma)) in 37 °C in a humid, 5% CO$_2$, 95% air (20% O$_2$) incubator or in hypoxic culture conditions (2.5% O$_2$) at 37 °C in a modular incubator chamber (Billups-Rothenberg) that was infused with a pre-analyzed gas mixture containing 2.5% O$_2$, 5% CO$_2$, and 92.5% N$_2$ (AirGas) or a cell culture incubator programmed for the desired O$_2$ tension. The input ASC numbers per well varied (~100 to ~3082 cells per well) depending upon post-sort cell counts. In MSC secretome media, the same number of ASC alone were cultured with factors for specified days. For MSC-free cultures or conventional media, RPMI with 10% FBS (R10) were used. Cells were harvested on designated days and IgG Elispot assays were performed[49]. The blood or BM ASC survival and function were assessed by Elispot assays, and their output values were expressed as the percentage of maximal IgG seceting ASC which typically occurred on days 1–3.

Exogenous factors included a variety of cytokines and growth factors, IL-5, IL-6, APRIL, BAFF, IFNg (R&D), IL-21 (Peprotech), bFGF (Thermofisher Scientific), and CXCL12 (Rockland) which were added to cultures at day 0 with ASC. The final concentrations of the added factors, which were optimized based on titrated experiments and the manufacturer's recommendations.

**Elispot assays.** To assess survival and Ig secretion function of cultured ASC, enzyme-linked immunospot (ELISpot) assay was used to quantify IgG-secreting cells in the cultures. These ELISpot assays used goat anti-human IgG for coating and alkaline phosphatase-conjugated goat anti-human IgG for detection[36,49]. Briefly, pre-wetted ELISpot plates (Millipore) were coated overnight at 4 °C with goat anti-human IgG (H + L) capture antibody (Novex/Fisher) or with BSA. Plates were then blocked with RPMI 1640 supplemented with 8% FBS and subsequently loaded with cultured ASCs and incubated at 37 °C for 16–18 h. Cells were removed and plates were washed six times. Bound Abs were detected using alkaline phosphatase-conjugated goat anti-human IgG secondary antibody (Jackson Immunoresearch). Spots were developed with ABC-AP Vector Blue Substrate reagents (Vector Laboratories). The spots were quantified using the Cellular Technology Limited; CTL (ImmunoSpot 5.0.9.21 software).

**BrdU cell proliferation assays.** Blood ASC cellular proliferation was evaluated on the basis of BrdU incorporation and using the BrdU Cell Proliferation Assay kit (Millipore). Briefly, BrdU (1:100 finally diluted, in BM secretome or media from Millipore's BrdU solution; 100 μL/well) was administered at the beginning (day 0)

of the cultures with the MSC secretome alone, secretome with sorted ASC cultures, or rapidly dividing non-irradiated MSC (used as positive controls). All cultures were then incubated at 37 °C in the air incubator (5% $CO_2$) for 48–72 h. All wells were gently washed twice with PBS by centrifuging at $500 \times g$ for 10 min at RT. The assay was performed in replicates.

**Proteomics**. Irradiated and non-irradiated BM-MSC or EV-depleted fractions pellets were lysed through end-to-end rotation at 4 °C for 45 min in RIPA buffer. The supernatant was transferred to new tubes. Proteins were reduced with 5 mM dithiothreitol (DTT) (56 °C, 30 min) and alkylated with 14 mM iodoacetamide (RT, 15 min in the dark). Detergent was removed by the methanol–chloroform protein precipitation method. Purified proteins were digested with 10 ng/μL Lys-C (Wako) in 50 mM HEPES pH 8.6, 1.6 M urea, 5% ACN at 31 °C for 16 h, then with 8 ng/μL Trypsin (Promega) at 37 °C for 4 h.

**Peptide purification and LC-MS/MS analysis**. A total of 0.5 μg of total protein per sample were used for proteomic analysis. Protein digestions were quenched by addition of trifluoroacetic acid (TFA) to a final concentration of 0.1%, followed by centrifugation to remove the precipitate. The peptides were desalted using a tC18 Sep-Pak cartridge (Waters) and lyophilized and subjected to LC-MS/MS analysis. Peptides were detected with a data-dependent Top20 method[50] in a hybrid dual-cell quadrupole linear ion trap—Orbitrap mass spectrometer (LTQ Orbitrap Elite, ThermoFisher, with Xcalibur 3.0.63 software). One full MS scan (resolution: 60,000) was performed in the Orbitrap at 10E6 AGC target for each cycle, and up to 20 MS/MS in the LTQ for the most intense ions were recorded. These sequenced ions were excluded from further analysis for 90 s. Precursor ions were required to have at least two charges for analysis. Maximum ion accumulation duration was 1000 ms for each full MS scan and 50 ms for MS/MS scans. All $MS^2$ spectra were searched using the SEQUEST algorithm (version 28)[51]. Spectra were matched against a database containing sequences of all proteins in the UniProt Human (*Homo sapiens*) database. We used the following parameters for database searching: 20 ppm precursor mass tolerance; fully digested with trypsin; up to three missed cleavages; fixed modification: carbamidomethylation of cysteine (+57.0214); variable modifications: oxidation of methionine (+15.9949). FDRs of peptide and protein identifications were evaluated and controlled to less than 1% by the target-decoy method[52] through linear discriminant analysis[53]. Peptides fewer than seven amino acid residues in length were deleted. We also applied a filter at the protein level to ensure the protein FDR is less than 1%.

**RNA transcriptomes of blood and BM ASC**. Paired end 100 bp reads were mapped to hg38, using Tophat2 to generate bam files. HTseq (version) was then used to assess read counts at the gene level in each gene. The mapped libraries were then normalized accounting for differences in size and dispersion using the default parameters of EdgeR for gene abundance levels. Read counts were converted to the log base 2 scale and principal component variance analysis (PCVA) in JMP-Genomics (version 8.0) was used to assess the contributions of Individual, Batch, Cell Population, and Tissue to the weighted average variance explained by the first three PCs. Since Batch explained 53% of the variance, in order to maximize the contrasts across the populations, we elected to use remove the batch effect using the SNM (supervised normalization of microarray) method[54] with Cell Population as the Biological variable and Batch as the adjustment variable with the Rm = True option. For differential expression analysis, we set a lower threshold of 3 log2 units, selected by plotting the coefficient of variance against average abundance. Lower-abundance features were removed for all downstream analyses. Differences among cell populations were assessed by analysis of variance on a gene-by-gene basis.

**Bioinformatics**. Fastq sequence files for the cell-sorted populations of four to six subjects were aligned to hg38 using hisat2 (v2.1.0) and HTSeq was used to generate gene-level counts which where normalized for library size and outlier transcript abundance with the TMM procedure in EdgeR. Resultant CPM (counts per million) were log2-transformed for each sample. Analysis of variance (ANOVA) was performed on these transcript abundance measures, contrasting the three ASC populations and three PC populations, retaining all genes at FDR < 0.05. From the original 17,640 genes, 2558 were found to be significantly differentially expressed between blood ASC and BM LLPC. Of these 2558 genes, 556 were shown to be potential interacting partners for the 91 BM-MSC secretome proteins using HIPPIE. These 556 genes were then standardized to account for difference in levels of expression across genes and hierarchically clustered using Ward's method in JMP Genomics (version 8.0). Furthermore, all potential PPI of YWHAZ and FN-1 (436 and 430, respectively) resulted in 126 overlapping potential PPI (of which 31 potential PPI are differentially expressed between blood ASC and BM LLPC). For both the set of 556 DEG and 126 overlapping genes, gene set enrichment analysis (GSEA) was performed using the *t*-statistic (derived from comparison of each gene between each pair of ASC populations) to rank genes, and pre-ranked gene set sets significantly enriched for high or low expression with FDR < 0.05 were deemed to be dysregulated between the two tissues. The expression of the standardized least-square-means (SLSM) for each gene in each gene set was then used to compute PC1 for the gene set, and these values were hierarchically clustered again using Ward's method in JMP Genomics.

**Statistical analyses**. Data are expressed as mean ± SD (standard deviation) and statistical differences between two groups (or two means of various experimental groups) and multiple groups were evaluated with Graphpad Prism 7 software (GraphPad Software). We used the JMP software version 8 to conduct an analysis of variance (ANOVA) using a Benjamini–Hochberg 5% false discovery rate to consider the interactions effects of treatment to analyze the data. Models were fit with day and treatment as fixed effects and significance of full model was reported.

**Data availability**. The proteomic data sets generated during the current study are available in the PeptideAtlas repository (Dataset Identifier: PASS01213; Password: RN7973ym), https://db.systemsbiology.net/sbeams/cgi/PeptideAtlas/PASS_View?identifier = PASS01213. The whole transcriptome RNA-seq data sets that support the findings of this study have been deposited in the public functional genomics data repository, NCBI GEO database (Accession Number: GSE116971) https://www.ncbi.nlm.nih.gov/geo/query/acc.cgi?acc = GSE116971. Any additional data that support the findings of this study are available from the corresponding author upon reasonable request.

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

## Acknowledgements

We thank Jennifer Scantlin, Claudine Nkurunziza, Anna Stephens, Hinel Patel, Maya Lindsay, Sonia Ros, and Sang N. Le for their dedication to human subject recruitment for this study. We also thank Raghavan Chinnadurai and Holly Lewis for his initial MSC lines; Wayne Harris and Ernestine Mahar in Department of Hematology and Medical Oncology; Aaron Rae in the Children's Healthcare of Atlanta & Emory University's Pediatric Flow Cytometry Core; and Robert E. Karaffa and Kametha T. Fife in the Emory Flow Cytometry Core (EFCC; one of the Emory Integrated Core Facilities (EICF)), for their assistance. We thank Vivien Warren for her technical assistance. We also thank the Atlanta Clinical & Translational Science Institute (ACTSI) for their assistance in obtaining BM aspirates. This work was supported by: NIH: NIAID: 1R01AI121252, R21AI094218, R21AI109601, 1P01AI125180, P01A1078907, R37AI049660, U01AI045969, HHSN266200500030C (N01-AI50029), U19AI109962, NIH/NCATS UL1 TR000454, R01DK109508.

## Author contributions

D.C.N. conceived the experimental design, carried out the experiments and wrote the manuscript. S.G. carried out the experiments and wrote the manuscript. H.X. and R.W. carried out the proteomics experiments. S.K. and I.A. helped with the experiments. J.G. helped develop the MSC isolation. K.-Y.C. and E.K.W. helped obtain the human BM aspirates. G.G. supervised the bioinformatics analysis. J.R. helped obtain human BM specimens. F.E.L. helped conceive the experimental design. T.D.R. and I.S. helped conceive the experimental design. F.E.-H.L. conceived the experimental design and wrote the manuscript.

## Additional information

**Competing interests:** F.E.-H.L. is the founder of MicroBplex, Inc., J.R. received grants from Stryker. A patent has been applied for by Emory University with F.E.L, I.S. and D.C. N. as named inventors. The patent application number is PCT/US2016/036650. The remaining authors declare no competing interests.

