## [Peer Review File · Nature Communications]

Reviewers' comments:

Reviewer #1 (Remarks to the Author):

The report characterises a system for promoting the survival of human antibody secreting cells that has as its critical component the supernatant from cultured bone marrow (BM) mesenchymal stromal cells. It is shown that either the MSC or supernatant from these cells can, with APRIL, sustain a fraction of peripheral blood ASC in vitro for periods of up to 50 days. The authors also show that making the cultures hypoxic, reflecting the bone marrow environment in which BM-resident ASC reside, that survival is further enhanced.

As far as it goes, this is interesting and an advance over current systems. On the negative side, however, there are no details of the mechanisms involved, either on the side of the factors produced by the stromal cells or the receptors and signaling pathways on the ASC side. In this regard, the report could be reduced to 2-3 figures, as that would incorporate the key findings.

I am a little puzzled with the variability of secretion that is reported, ranging from 200 ng/mL IgG at 28 days for 3000 ASC, or 2000 ASC for 21 days or 1000 ASC for 20 days (5a). This seems very odd, given the small errors associated with each measurement. Some kind of table indicating the variation associated with different donors would be helpful.

Equally, the description of the methodology could be improved, particularly the description of the BrdU incorporation experiment, which basically now says we followed the manufacturer's instructions. Unfortunately, without positive controls (eg, a cell line or in vitro stimulated B cells) it is impossible to interpret the significance of an OD value of 0.12. Also, this assay would be far more informative over a time course as it does appear that there is increasing proliferation on days 2 to 3.

I am also a little surprised at the significance of some of the differences that are described. In particular, the data in Fig 4c at day 14 for IgG hypoxia and normoxia with APRIL are described in the text as being significantly different at $p < 0.0001$ and yet the standard deviations are almost overlapping (222 +/- 31 versus 184 +/- 22). Is this correct? Although I do note that there is no indication in any figure or text of exactly what is plotted in the figures (mean +/- SD? +/- SE?). That would be helpful. In keeping with this statistical theme, there are no errors or variation indicated in Fig 5a-c or Fig 1e.

In conclusion, there is little wrong with the report, but it is not the most exciting thing I have read. For example, the single cell memory culture appears to be very similar to that of Lanzaveccia, in terms of mitogens and culture conditions for supporting plasma cells. He has not characterised its potential, nor did he join the memory and ASC cultures together as done here, so that is innovative.

Reviewer #2 (Remarks to the Author):

Taking their cues from published work on factors that improve plasma cell survival in vitro and in vivo, Nguyen et al. look at factors that might improve human antibody secreting plasma cell (ASC) survival in vitro. This poor survival is a serious limitation to our ability to study this extremely important but rare cell type, and advances in culture systems for ASC would be very welcome and of broad interest to immunologists. However, the data presented here do not appear to be generated (or perhaps described) in a rigorous enough way to indicate that a truly novel method for in vitro prolongation of ASC life has been discovered. While there is certainly potential here, as indicated by the culturing of blood-derived antigen-specific ASC from vaccinated, infected and SLE patients, more attention to controls and quantification is required before this work is ready for

publication.

Comments/concerns:

1) There are several problems with the presentation and interpretation of the data in Figure 1 (and continued through the data). In the graph in Fig 1a, I believe the Y-axis label is incorrect. 250% survival is a meaningless notion. It should be labelled either total cell number or cell expansion relative to the starting number, but survival >100% is not possible. Using total ASC number would enable us to know if there had been cell expansion or death, as the numbers over time could be related to the total # of sorted ASC placed into the cultures.

The notions of "restored IgG secretory function of cultured ASC" and "a stunned non-secretory state" needs some explaining (and proving). This assumes that some of the cells put into the culture were non-functional phenotypic ASC, but this should be clarified by doing an ELISpot directly on input cells at the start of the culture and relating spot # to total sorted cells put into the cultures. At no point in Fig 1 do the investigators show cell numbers over time, but only show %.

Cell number is a critical parameter and should be shown for each time point. Later, Fig 1a is referred to: "As shown in Fig 1a, IgG ELISpot frequencies of co-cultured ASC increased greatly during the first 24 hours of culture (250-500%", but no 500% (or 5-fold) increase is evident in the Fig 1a data.

For Fig 1c, % IgG ASC survival is shown, but how were the sorted ASC cultured? Were supporting cells added back after the sort? If not, is it possible that the lack of supporting cells, rather than simple sorting, contributed to ASC loss? This should be determined.

2) How were the BM derived MSC prepared to exclude any ASC co-purifying with the MSC, as these two cell types can intimately associate, as the authors acknowledge? A third sample should have been included in Fig 1a, which was MCS alone, to formally exclude ASC co-purifying with the MCS. Also, how were the ASC identified in the image of Fig 1b? Simple morphology? If the image is to be considered as part of the data, quantitation and statistics should be included.

3) The MSC secretome needs better definition. The Methods suggest it is most like an MSC conditioned medium. Was it spun at high speed or filtered? Irradiated cells or not? This is important to help understand the synergy between April and the MSC secretome in supporting ASC in vitro.

4) This conclusion "a significant recovery of ASC function was observed within 24 hours" (line 176) cannot be supported without knowledge of cell numbers. It was shown in Fig 1d that ASC from recently vaccinated individuals are dividing.

5) While hypoxia modestly improves ASC survival under some conditions, statistical analyses of the survival data between the groups in Fig 4c and d is required. Also, the relationship between the number of spots shown in Fig 4b is not consistent with the %IgG ASC survival unit in the graph in 4c.

6) In Fig 5, the ASC # and output (vaccine-specific IgG) for the cells at the start of the cultures (in the first few days) should have been included, in order to understand what fraction of that response was still present after long-term culture in the conditions described here.

Minor points:

a) The IgG concentrations listed in the abstract are irrelevant as there is no accompanying information about cell number or culture volume. These details do not belong in the abstract; the general statement about that ability to detect secreted IgG is adequate.

b) There are a number of grammatical errors/typos, mostly relating to singular or plural terms

c) Line 305: The authors cannot conclude that "Again, these numbers are consistent with a highly efficient single antigen-specific ASC response after vaccination" as their earlier data (Fig 1d)

indicates that the antigen-specific ASC are dividing post-vaccination, so each well may not contain a single cell.

Reviewers' comments:

Reviewer #1 (Remarks to the Author):

1. The report characterises a system for promoting the survival of human antibody secreting cells that has as its critical component the supernatant from cultured bone marrow (BM) mesenchymal stromal cells. It is shown that either the MSC or supernatant from these cells can, with APRIL, sustain a fraction of peripheral blood ASC in vitro for periods of up to 50 days. The authors also show that making the cultures hypoxic, reflecting the bone marrow environment in which BM-resident ASC reside, that survival is further enhanced.

As far as it goes, this is interesting and an advance over current systems. On the negative side, however, there are no details of the mechanisms involved, either on the side of the factors produced by the stromal cells or the receptors and signaling pathways on the ASC

side. In this regard, the report could be reduced to 2-3 figures, as that would incorporate the key findings.

We appreciate the reviewer's acknowledgement that this system is "interesting and an advance over current systems". To address the concerns of detailed mechanisms, we have added new proteomics data from the stromal cells combined with receptor and signaling pathways from the transcriptome analysis of blood and BM ASCs in figure 4. This new figure highlights the protein-protein interactions of the MSC secretome and the maturing ASC in the presence of the in vitro BM microniche mimic. We identify 91 novel proteins that are likely involved in the ASC survival mechanisms. We further validate 2 of the top protein candidates, fibronectin and YWHAZ (14-3-3 protein) (see **figure 5**).

Interestingly, low abundant cytokines such as IL-6 were not detected by proteomics due to their low abundance (Sze et al). However, we demonstrate inhibition with anti-IL-6 antibodies is critical for ASC survival (see **figure 3e**).

2. I am a little puzzled with the variability of secretion that is reported, ranging from 200 ng/mL IgG at 28 days for 3000 ASC, or 2000 ASC for 21 days or 1000 ASC for 20 days (5a). This seems very odd, given the small errors associated with each measurement. Some kind of table indicating the variation associated with different donors would be helpful.

We agree that we needed to provide improved ASC secretion per day. Due to space limitation of this manuscript and new mechanistic data, we have removed this figure to provide more detailed kinetics assays in another manuscript. To address the mechanistic questions, we added **figure 4 and 5**.

3. Equally, the description of the methodology could be improved, particularly the description of the BrdU incorporation experiment, which basically now says we followed the manufacturer's instructions. Unfortunately, without positive controls (eg, a cell line or in vitro stimulated B cells) it is impossible to interpret the significance of an OD value of 0.12. Also, this assay would be far more informative over a time course as it does appear that there is increasing proliferation on days 2 to 3.

We have expanded the methods and as well as published a recent manuscript (Nguyen D et al JEV, 2018) to describe the methodology. For the BrdU experiments, we added a new figure using proliferating MSC as a control (**figure 2c**). The non-irradiated MSC proliferate (with increased cell numbers in our cultures) and readily incorporate BrdU in culture. As additional controls, we added the MSC secretome which is negative (indicating no contaminating cells). We expected this result since the MSC secretome has been filtered in 0.2 micron filters after collection and prior to adding to the ASC cultures (see methods). Background of non-irradiated MSC without BrdU is also added as a negative control. In **figure 2c**, there is no BrdU uptake in ASC + secretome demonstrating no proliferation of the ASC. Figure 2c is representative of 3 separate experiments.

4. I am also a little surprised at the significance of some of the differences that are described. In particular, the data in Fig 4c at day 14 for IgG hypoxia and normoxia with

APRIL are described in the text as being significantly different at $p < 0.0001$ and yet the standard deviations are almost overlapping (222 +/- 31 versus 184 +/- 22). Is this correct? We have now repeated this experiment from long-lived vaccine (tetanus) blood ASC from 4 separate healthy adults (now **figure 6**). We removed the healthy donor subject and performed statistics using the JMP software version 8 to conduct an analysis of variance (ANOVA) using a Benjamini-Hochberg 5% false discovery rate to consider the interactions effects of treatment to analyze the data. Models were fit with day and treatment as fixed effects and significance of full model was reported (See methods, page 17). Standard deviations are also very small on most of the days such that they are not appreciated beyond the symbol of the figure legends. Although there were little at early time points, the divergence of the survival curves are highly statistically significant between the normoxia and hypoxia conditions in the Secretome and Secretome + APRIL conditions reflective in the p-values (below the figure).

5. Although I do note that there is no indication in any figure or text of exactly what is plotted in the figures (mean +/- SD? +/- SE?). That would be helpful. In keeping with this statistical theme, there are no errors or variation indicated in Fig 5a-c or Fig 1e.

Error bars are based on standard deviation and some were so small that the symbols are bigger than the error bars. Error bars based on standard deviations are included and p-values for each of the graphs are shown in all the figures.

6. In conclusion, there is little wrong with the report, but it is not the most exciting thing I have read. For example, the single cell memory culture appears to be very similar to that of Lanzavecchia, in terms of mitogens and culture conditions for supporting plasma cells. He has not characterised its potential, nor did he join the memory and ASC cultures together as done here, so that is innovative.

We appreciate the reviewer's positive opinion of the method that is novel in this paper. To address the other major concerns of mechanisms, we have included **figure 4 & 5** to demonstrate the components of the MSC secretome and to validate the importance of novel pathways that are differentially regulated in by a combination of the MSC secretome and the hypoxic microenvironment.

We removed the single cell in vitro differentiated memory B cells cultures for the additional mechanistic studies which are quite novel and exciting. However, we disagree that our method is very similar to Lanzavecchia's approach. Others may have proliferated single memory B cell cultures with similar mitogens; however, it is our group to first use a combination of cell-free system of single memory B cell for proliferation and differentiation (for 6 days) and then a sequential 6-day culture to collect IgG secretion in the novel BM microniche mimic system to measure enough specific antibodies by ELISA. We will provide these methods in a follow up manuscript.

For this manuscript, we emphasized the in vitro BM mechanisms of LLPC maturation by a novel integrated MSC proteomics and BM and blood ASC transcriptomics analysis to understand maturation and survival of LLPC in the BM microniche.

Reviewer #2 (Remarks to the Author):

Taking their cues from published work on factors that improve plasma cell survival in vitro and in vivo, Nguyen et al. look at factors that might improve human antibody secreting plasma cell (ASC) survival in vitro. This poor survival is a serious limitation to our ability to study this extremely important but rare cell type, and advances in culture systems for ASC would be very welcome and of broad interest to immunologists. However, the data presented here do not appear to be generated (or perhaps described) in a rigorous enough way to indicate that a truly novel method for in vitro prolongation of ASC life has been discovered. While there is certainly potential here, as indicated by the culturing of blood-derived antigen-specific ASC from vaccinated, infected and SLE patients, more attention to controls and quantification is required before this work is ready for publication.

We appreciate the reviewer's comments and have revised the presentation of the data that may not have been described with enough rigor. We now have performed many experiments to address these concerns.

Survival of highly activated cells after high pressure FAC sorting has been challenging to all immunologist; however, we do not make claims to improving cell sorting methods. In this revised manuscript, we actually provide novel mechanisms of a new cell-free in vitro culture system that mimics the BM microniche to maintain survival of ASC AFTER sorting. We now describe specific factors of the BM microniche that are involved in the mechanisms of survival of blood ASC for months in vitro. **The new figure 2** shows valuable controls that address many issues that were raised. **Figure 4 & 5** provide novel mechanisms provided by the BM microniche.

Comments/concerns:

1) There are several problems with the presentation and interpretation of the data in Figure 1 (and continued through the data). In the graph in Fig 1a, I believe the Y-axis label is incorrect. 250% survival is a meaningless notion. It should be labelled either total cell number or cell expansion relative to the starting number, but survival >100% is not possible. Using total ASC number would enable us to know if there had been cell expansion or death, as the numbers over time could be related to the total # of sorted ASC placed into the cultures.

We agree with the reviewer and changed all the figures to provide the maximal number of IgG ASC cells typically found on day 1-3 of the cultures. The maximal number is normalized to 100% of the ASC. Since they do not proliferate (**figure 2c**), we put the same number of cells in each well that was counted by the FAC sorter and used the IgG Elispot numbers to assess for survival and function which can be variable from subject to subject. To address the reviewer's concerns, we compared the FAC sorter counts and manual counting of the cultures 4-6 hours after cells settle in the well, we found the manual numbers to varied from 20-70% of the FACS counts. Thus, we used the

percentage of maximal IgG Elispots on day 1-3 and measured number of remaining Elispots on subsequent days.

The number of IgG Elispots numbers on day 0, typically ranged from 15-30% of the total FACS counted input numbers. This low number may be due a multitude of the following factors: unmeasured ASC secreting IgA in the cultures which can make up 2-5X the numbers, overestimates of the post-FAC sort numbers due to dead or dying cells that are counted by the sorter but are not measured by Elispot assays, or cells that are counted as an event by the FACS machine but are not viable as we demonstrate by the high pressure stress (**figure 2b**). We measured purity of the ASC which ranged from 85 to 95% which does not contribute to the low numbers. Thus to measure survival and function of the ASC in our in vitro cultures conditions, we normalized the survival with the maximal number IgG ASC by Elispot detected on day 1-3 to measure % survival of the cells in the cultures since no further proliferation occurred in the ASC cultures. Additionally, in this manuscript, we make no claims of improving survival during FAC sorting. Our results demonstrate enhanced survival of ASC AFTER the sorting process and are maintained in our cell-free BM mimic for > 50 days. Other early ASC found in the lymph nodes may indeed be proliferating but our assays were performed on human ASC circulating in the blood and not from lymph nodes and the circulating human ASC are not actively proliferating by their lack of incorporation of BrdU despite Ki67+ staining.

The notions of "restored IgG secretory function of cultured ASC" and "a stunned non-secretory state" needs some explaining (and proving). This assumes that some of the cells put into the culture were non-functional phenotypic ASC, but this should be clarified by doing an ELISpot directly on input cells at the start of the culture and relating spot # to total sorted cells put into the cultures. At no point in Fig 1 do the investigators show cell numbers over time, but only show %. Cell number is a critical parameter and should be shown for each time point.

We performed Elispot assays immediately after FAC sorting. Day 0 Elispots are the numbers of IgG ASC secretors directly after FAC sorting. We found that some of the cells are revived from a stunned state after FAC sorting on day 1-3 as shown in **figure 1, 2** thus we used the maximal numbers of IgG ASC possible in the cultures are detected on day 1-3. The same number of input cells are placed into multiple replicate wells under the specified culture conditions for each experiment. Since the cells were not proliferating (as shown with the BrdU experiments), we were measuring survival. The input post-sort cell numbers range from 100-3,000 cells placed in each well. Although the input cell number is critical at each time point, with such low numbers of cells (less than 100 to 3,000 cells per well) enumerating viable cells after the cultures is very difficult; thus, to enumerate the IgG ASC numbers, we measured the frequencies by single cell IgG Elispots assays.

Although FAC sorting affects viability, to maximize purity of ASCs, a relatively rare cell type after ficoll separation, we employed FAC sorting (85-95% purity) vs other methods magnetic bead selection.

Later, Fig 1a is referred to: " As shown in Fig 1a, IgG Elispot frequencies of co-cultured

ASC increased greatly during the first 24 hours of culture (250-500%", but no 500% (or 5-fold) increase is evident in the Fig 1a data.

We did find that IgG secretion is affected by high pressure FAC sorting since we could only detect 25 to 40% on day 0 (immediately after sorting) compared to the maximal IgG numbers on day 1-3. This correction was made for each of the experiments.

For Fig 1c, % IgG ASC survival is shown, but how were the sorted ASC cultured? Were supporting cells added back after the sort? If not, is it possible that the lack of supporting cells, rather than simple sorting, contributed to ASC loss? This should be determined.

This experiment is now described in **figure 2b** in the revised manuscript. In this experiment, total PBMC were isolated and divided into four wells equally. There are no cells added back. For all the conditions, the same number of total PBMC were collected after each condition. The first condition, no stain or sort (untouched PBMC), was just plated for Elispots. The second condition took the same number of PBMC and added antibodies and washed and performed Elispots on the total PBMC. The third condition took the same number of PBMC and put them through the rigorous process of FAC sorting and recovered all the cells. Then, Elispots were performed. The last condition added antibodies and put them through the FAC sorter and also recovered ALL the cells. Again, Elispots were performed. Despite similar numbers of PBMC in each condition, the numbers of IgG Elispots varied tremendously demonstrating a functional loss of 60-75% with the high pressure FAC sorting process. Thus, we concluded the high pressure sorting process affects viability and functionality of the ASC resulting in 25-40% of the Elispot numbers on day 0 compared to day 1-3. This is clarified in the text (page 6).

2) How were the BM derived MSC prepared to exclude any ASC co-purifying with the MSC, as these two cell types can intimately associate, as the authors acknowledge?

We appreciate these concerns and added additional controls in **figure 2d**. Additional details are described in the method on page 14. Briefly, the MSC secretomes were generated from MSC derived from BM adherent cells. MSC were passaged typically 7-8 divisions. The MSC were irradiated and plated with supernatants collected daily for 7 days. Each lot of MSC secretome was filtered through a 0.2 micron filter; removing any contaminating cells and debris in the MSC secretome preparations prior to adding to the FAC sorted ASC cultures. The controls of iMSC cultures (after 7-8 passages which is often 4-6 weeks after ficoll separation of the primary BM aspirate) or the MSC secretomes showed no detectable IgG Elispots (**figure 2d**).

A third sample should have been included in Fig 1a, which was MCS alone, to formally exclude ASC co-purifying with the MCS.

We appreciate these comments. This experiment is included in **figure 2d** in the revised manuscript.

Also, how were the ASC identified in the image of Fig 1b? Simple morphology? If the image is to be considered as part of the data, quantitation and statistics should be included.

We performed flow staining on the BM MSC cultures after passage 5 and found the cultures were nearly all (>90%) of the cells are CD73+CD90+CD45- (unpublished results) illustrating very few if any contaminating ASC. Furthermore, after multiple passages, the MSC alone are completely negative in IgG Elispot assays demonstrating no contaminating ASC attached to the primary BM MSC. Additionally, we show that BM MSC and the MSC secretome are equivalent in supporting survival of ASC. Because we focus on the MSC secretome rather than iMSC co-cultures for the remainder of the manuscript, we removed the morphology (old figure 1b) due to space limitations and to provide additional mechanistic data **figure 4 & 5**.

3) The MSC secretome needs better definition. The Methods suggest it is most like an MSC conditioned medium. Was it spun at high speed or filtered? Irradiated cells or not? This is important to help understand the synergy between April and the MSC secretome in supporting ASC in vitro.

Additional details are described in the method on page 14. This is also described in point 2.

4) This conclusion "a significant recovery of ASC function was observed within 24 hours" (line 176) cannot be supported without knowledge of cell numbers. It was shown in Fig 1d that ASC from recently vaccinated individuals are dividing.

We respectfully disagree that Ki67 means "actively proliferating cells". Ki67+ cells definitely includes actively proliferating cells but can also include recently proliferated cells because Ki67+ cells can be in G1, S, G2, and M phase. Others have shown that cells co-stained with Ki67 and BrdU identified many cells that are Ki67+BrdU- but only a subset of the Ki67+ BrdU+ (cells in S phase) (Tanaka R et al 2011 Journal of Histochemistry & Cytochemistry 59(8) 791–798), Massey A Plos One 2015). Thus, cells actively undergoing proliferation are Ki67 and BrdU positive whereas Ki67 positive cells include both actively proliferating cells and those that have recently proliferated.

Our circulating ASC are positive for Ki67 (>90%), but we find that they do not incorporate BrdU demonstrating that they are not in S-phase or actively proliferating. Thus, in a culture of non-proliferating blood ASCs, we plate the same number of ASC in the replicate wells at the start of the culture. Since they are not proliferating, we show that we are actually measuring cell survival.

5) While hypoxia modestly improves ASC survival under some conditions, statistical analyses of the survival data between the groups in Fig 4c and d is required. Also, the relationship between the number of spots shown in Fig 4b is not consistent with the %IgG ASC survival unit in the graph in 4c.

In response to point 4 by reviewer 1, we have now repeated this experiment from long-lived vaccine (tetanus) blood ASC from 4 separate healthy adults. This is now the new **figure 6**. We removed the healthy donor subject and performed statistics using JMP software version 8 to conduct an analysis of variance (ANOVA) using a Benjamini-Hochberg 5% false discovery rate to consider the interactions effects of treatment to analyze the data. Models were fit with day and treatment as fixed effects and significance of full model was reported. The divergence of the survival curves is highly statistically

significant between the normoxia and hypoxia conditions in the secretome only and secretome + APRIL cultures (**figure 6a**).

6) In Fig 5, the ASC # and output (vaccine-specific IgG) for the cells at the start of the cultures (in the first few days) should have been included, in order to understand what fraction of that response was still present after long-term culture in the conditions described here.

We have cultures from day 0, 1, 3, 5, 7, and have measured the total IgG and vaccine-specific IgG with the Elispots numbers from each one. They sequentially increase in cumulative antibody concentrations within the supernatants on each subsequent day. However, we removed this figure to include **figure 4 & 5** novel mechanisms of the factors in the MSC secretome and the pathways involved.

Minor points:

a) The IgG concentrations listed in the abstract are irrelevant as there is no accompanying information about cell number or culture volume. These details do not belong in the abstract; the general statement about that ability to detect secreted IgG is adequate.

This has been removed.

b) There are a number of grammatical errors/typos, mostly relating to singular or plural terms

These errors have been corrected.

c) Line 305: The authors cannot conclude that "Again, these numbers are consistent with a highly efficient single antigen-specific ASC response after vaccination" as their earlier data (Fig 1d) indicates that the antigen-specific ASC are dividing post-vaccination, so each well may not contain a single cell.

This figure has been removed.

REVIEWERS' COMMENTS:

Reviewer #1 (Remarks to the Author):

Many of the issues of the first version of the manuscript have been addressed. I note the authors are still not presenting ASC numbers but have instead chosen to use maximal frequency as 100% and normalise everything to this. I would have thought numbers would be easier but this is probably OK.

I remain somewhat confused by the idea of stunned ASC due to sorting, as this is not something others who sort ASC have reported, but I am prepared to accept their suggestion although I think it could have been more vigorously examined, even by microscopic examination of the cells, or intracellular Ig staining.

Of most concern to me, which I think the authors should address in their discussion is that the effect of APRIL on survival in the presence of the secretome is in almost every case restricted to the first 1-3 days. After this time, the decay in ASC is almost identical with or without APRIL. For example, Fig 3c tetanus, tetanus, MMR and Fig 3d. The curves are essentially identical after the maximal point inflexion. Doesn't this suggest strongly that the effect of APRIL is predominantly if not only in the recovery phase after sorting?

Secondly, there should be some explanation of the inhibition data. Three antibodies are used, specific for IL6, FN1 and YWHAZ that collectively reduce survival by about 150% on d3. Clearly this can't be the case, suggesting some overlap, redundancy or something else. While it is curious the authors have never tried these together, they might discuss what the collective effect might be.

Finally, I am a little concerned by the numbers involved. The authors add a few thousand sorted ASC into wells containing tens of thousands of MSC. Yet my impression is that in the bone marrow, this ratio is likely to be reversed. This might be commented upon as well.

Reviewer #2 (Remarks to the Author):

My major concerns with the original manuscript by Nguyen et al. centered on the importance of quantification of plasma cells (ASC) in their assays and cultures, and on the need for important controls for their co-cultures, because both ASC and the potential supporting cells they describe are very small subpopulations in blood, lymphoid organs and bone marrow. It would be easy to include unintentional contaminants in such cultures.

The authors have gone to great lengths in this revised manuscript to redefine the ASCs using functional assays, and also have included controls that alleviate the concern about contamination of cell populations. While some aspects of the ASC culture system are not novel (April's role, as acknowledged by the authors), the careful work performed here now represents a valuable step forward in the manipulation of this rare and critical cell type. Recommended for publication.

Reviewer #1 (Remarks to the Author):

Many of the issues of the first version of the manuscript have been addressed. I note the authors are still not presenting ASC numbers but have instead chosen to use maximal frequency as 100% and normalise everything to this. I would have thought numbers would be easier but this is probably OK.

We appreciate the reviewer's comments.

I remain somewhat confused by the idea of stunned ASC due to sorting, as this is not something others who sort ASC have reported, but I am prepared to accept their suggestion although I think it could have been more vigorously examined, even by microscopic examination of the cells, or intracellular Ig staining.

We have actually examined the cells under the microscope after FACS sorting and the cells appear variable. Typically, there are decreased numbers of them (~40-80% compared to the cell numbers reported by FACS counting). Intracellular staining may have been a little difficult technically since some blood ASC also contain surface Ig expression and thus may not distinguish from any B cells.

Of most concern to me, which I think the authors should address in their discussion is that the effect of APRIL on survival in the presence of the secretome is in almost every case restricted to the first 1-3 days. After this time, the decay in ASC is almost identical with or without APRIL. For example, Fig 3c tetanus, tetanus, MMR and Fig 3d. The curves are essentially identical after the maximal point inflexion. Doesn't this suggest strongly that the effect of APRIL is predominantly if not only in the recovery phase after sorting?

We appreciate the reviewer's comment and recalculated the slopes of the ASC survival after the maximal point of inflexion for the MSC secretome alone and the MSC secretome with APRIL. There were some samples with different slopes but from the 5 samples, we were unable to ascertain significant differences. The lack of differences may be due to two interpretations: (1) the small sample size and

lack of biological replicates and time points to detect significance or (2) as the reviewer suggests: the effect of APRIL is primarily in the recovery phase. A sentence was included in the manuscript on page 6.

Secondly, there should be some explanation of the inhibition data. Three antibodies are used, specific for IL6, FN1 and YWHAZ that collectively reduce survival by about 150% on d3. Clearly this can't be the case, suggesting some overlap, redundancy or something else. While it is curious the authors have never tried these together, they might discuss what the collective effect might be.

We appreciate the reviewer's comment about the potential redundancy of function of these 3 proteins since the collective inhibition would be greater than 100%. At present, we do not have data using all 3 antibodies together. Nonetheless, we have added a sentence in the discussion to reflect the possible overlap or redundancy of function from these 3 proteins (page 11).

Finally, I am a little concerned by the numbers involved. The authors add a few thousand sorted ASC into wells containing tens of thousands of MSC. Yet my impression is that in the bone marrow, this ratio is likely to be reversed. This might be commented upon as well.

We agree with the reviewer's comments that the ratio of the BM ASC and the BM MSC may be closer to 1:1 or 10:1. Previous reports by Castro-Malaspina H et al Blood 1980 report frequencies of BM MSC as 1:10,000 or 1:100,000 BM mononuclear cells (BMNC) while we report the BM LLPC may be 5:10,000 of the BMNC (or for all BM ASC 2:1,000 BMNC) (Halliley JL 2015 et al). We initially tested ratios of ASC:MSC of 1:1, 1:2, 1:3, 1:5, 1:8, 1:10, 1:12, 1:15, 1:20, and 1:25, and found that 1:10 provided the best ASC support. Due to limited fresh human ASC numbers for each experiment, we only varied the MSC frequencies. However, in these ASC: MSC ratio experiments, we used the same volume of media. We now know that the MSC secretome is critical for the ASC survival and believe the number of MSC per volume and days of collection may be more important than the ratio of MSC and ASC to mimic the BM microniche. A sentence in the discussion was added to address this point (page 5).

Reviewer #2 (Remarks to the Author):

My major concerns with the original manuscript by Nguyen et al. centered on the importance of quantification of plasma cells (ASC) in their assays and cultures, and on the need for important controls for their co-cultures, because both ASC and the potential supporting cells they describe are very small subpopulations in blood, lymphoid organs and bone marrow. It would be easy to include unintentional contaminants in such cultures.

The authors have gone to great lengths in this revised manuscript to redefine the ASCs using functional assays, and also have included controls that alleviate the concern about contamination of cell populations. While some aspects of the ASC culture system are not novel (April's role, as acknowledged by the authors), the careful work performed here now represents a valuable step forward in the manipulation of this rare and critical cell type. Recommended for publication.

We appreciate the reviewer's comments and review.